# Study on the Mechanism of Solid-Phase Oxidant Action in Tribochemical Mechanical Polishing of SiC Single Crystal Substrate

**DOI:** 10.3390/mi12121547

**Published:** 2021-12-12

**Authors:** Wanting Qi, Xiaojun Cao, Wen Xiao, Zhankui Wang, Jianxiu Su

**Affiliations:** School of Mechanical and Electrical Engineering, Henan Institute of Science and Technology, Xinxiang 453003, China; qwt245147@163.com (W.Q.); cxj18303659239@163.com (X.C.); xw68886@163.com (W.X.); luckywzk@126.com (Z.W.)

**Keywords:** 6H-SiC, fixed abrasive, tribochemical mechanical polishing, solid-phase oxidant, dry polishing

## Abstract

Na_2_CO_3_—1.5 H_2_O_2_, KClO_3_, KMnO_4_, KIO_3_, and NaOH were selected for dry polishing tests with a 6H-SiC single crystal substrate on a polyurethane polishing pad. The research results showed that all the solid-phase oxidants, except NaOH, could decompose to produce oxygen under the frictional action. After polishing with the five solid-phase oxidants, oxygen was found on the surface of SiC, indicating that all five solid-phase oxidants can have complex tribochemical reactions with SiC. Their reaction products are mainly SiO_2_ and (SiO_2_)x. Under the action of friction, due to the high flash point temperature of the polishing interface, the oxygen generated by the decomposition of the solid-phase oxidant could oxidize the surface of SiC and generate a SiO_2_ oxide layer on the surface of SiC. On the other hand, SiC reacted with H_2_O and generated a SiO_2_ oxide layer on the surface of SiC. After polishing with NaOH, the SiO_2_ oxide layer and soluble Na_2_SiO_3_ could be generated on the SiC surface; therefore, the surface material removal rate (MRR) was the highest, and the surface roughness was the largest, after polishing. The lowest MRR was achieved after the dry polishing of SiC with KClO_3_.

## 1. Introduction

As a third-generation semiconductor material, SiC has excellent chemical and physical properties [1,2] and widely used in satellite communications, integrated circuits and consumer electronics [3,4,5]. However, SiC is characterized by high hardness, high brittleness, and good physical and chemical stability, therefore, it is typically a difficult material to machine [6].

At present, the common method for ultra-smooth processing of SiC is free abrasive chemical mechanical polishing (CMP), which achieves the ultra-smooth, damage-free, and ultra-flat surface processing of the workpiece through acombination of chemical etching of polishing solution and mechanical action of abrasive, and is one of the more effective global flattening processing methods for semiconductor materials [7,8]. However, CMP has the following disadvantages, low processing efficiency, poor environmental friendliness, poor surface consistency, and poor process engineering controllability [9]. Fixed abrasive chemical machining can effectively avoid the above disadvantages of free abrasive machining and has become one of the emerging technologies in the field of ultra-precision machining [10,11,12]. Fixed abrasive tribochemical mechanical polishing is a fixed abrasive chemical-mechanical processing technology, which can use the abrasive and chemical additives in the polishing pad and the surface of the workpiece in a tribochemical reaction to change the surface of the workpiece material and chemical organization. This mechanism achieves the efficient removal of its material; therefore, is the process increasingly gaining the attention of researchers [13].

The core of fixed abrasive tribochemical mechanical polishing is the tribochemical reaction produced on the surface of the workpiece in the polishing process. Therefore, understanding its chemical reaction mechanism is the key to studying its processing mechanism. Usually, there are two kinds of tribochemical reaction, one is the tribochemical reaction in the dry friction state, and the other is the tribochemical reaction in the lubricated state [14]. Tribochemical mechanical polishing does not use apolishing solution containing free abrasive, but instead uses tribochemical mechanical polishing tools. The polishing slurry is trace deionized water, or chemical solution, without adding any free abrasive, its tribochemical reaction is mostly wet tribochemical action, therefore, the workpiece material removal method is mostly tribochemical wear removal [15]. Since the workpiece obtained by this method has the advantages of low residual stress, flatness, and ultra-smoothness without damage, it has gained the attention of a large number of researchers. Z. Zhu et al. [16] conducted tribochemical mechanical polishing tests on SiC using abrasive-free oxidants H_2_O_2_, CrO_3_, and KMnO_4_, respectively, and excellent surfaces with surface roughness less than 50 nm and residual stress less than 50 MPa were obtained. S. Kitaoka et al. [17] proposed the theory of tribochemical wear based on a hydrothermal reaction for SiC and Si_3_N_4_ anaerobic ceramics. It is believed that SiC undergoes a tribochemical reaction with water at 120 °C and a SiO_2_ reaction layer is generated on its surface, which achieves ultra-precision machining after the removal of abrasive particles. Yusuke Ootani et al. [18] studied the kinetics of the tribochemical reactions of Si_3_N_4_ and SiC under an aqueous environment and analyzed different tribochemical reaction mechanisms during the lapping of Si_3_N_4_ and SiC. Zhou F et al. [19] proposed the wear mechanism of SiC/SiCin water related to the microfracture of the ceramic and the instability of the tribochemical reaction layer.

In summary, the current tribochemical mechanical polishing for SiC is mainly used in water-based polishing solutions for tribochemical mechanical polishing, which contains chemical additives in the polishing slurry, and the discharge of the polishing slurry will bring about environmental pollution and an increase in processing costs. Therefore, the use of fixed abrasive dry tribochemical mechanical polishing can reduce its chemical pollution and production cost. However, there are few studies on the dry tribochemical mechanical polishing of fixed abrasives regarding SiC, and the mechanism of its action arestill not clear enough, particularly the oxidation mechanism of the dry friction of SiC workpieces is still unclear and needs further research and exploration.

In this paper, five solid-phase oxidants were selected for dry polishing with a 6H-SiC single crystal substrate.The changes to their 3D morphology, compounds, and elements on the workpiece surface before and after polishing were examined to analyze their surface tribochemical reactants and to study the oxygen production mechanism of their solid-phase oxidants during the tribochemical mechanical polishing of the fixed abrasive. The study can provide help to understand the mechanism of oxygen production and the tribochemical reaction mechanisms of SiC during fixed abrasive tribochemical mechanical polishing, providing aid regarding the selection of a solid-phase oxidant for the fixed abrasive tribochemical mechanical polishing pad and the formulation of a solid-phase oxidant for a fixed abrasive.

## 2. Experiment and Characterization

### 2.1. Polishing Test

The test sample was the n-type of a 6H-SiC single crystal substrate (Tianke Heda, Beijing, China) with a thickness of 0.4 mm, a diameter of 50.8 mm, and an initial surface roughness of 6–7 nm. The SiC was pasted to the center of the carrier table with paraffin wax before the test, and was dry polished on the C side (0001) at room temperature using the ZYP230 rotary oscillating gravity lapping and polishing machine (Kemai, Shenyang, China). The processing principle is shown in Figure 1, and the polishing process parameters are shown in Table 1. The polishing pad used for the test was a polyurethane polishing pad, and the polishing medium was five typesof solid-phase oxidant. The specific compositions are shown in Table 2.

The solid-phase oxidant was spread evenly on the polishing pad, as shown in Figure 2a, and the dry polishing process had a dosing rate of 20g/h. Figure 2b,c show the beginning of the dry polishing process, during the dry polishing process, and after the completion of dry polishing, respectively. The single-factor method was used for the experiments and analysis to explore the oxygen production mechanisms of different solid-phase oxidant polishing.

The mass of each sample was measured using a precision electronic balance.Before and after its processing, the difference was calculated, and the material removal rate (MRR, nm/min) for polishing was calculated using Equation (1). The surface roughness and 3D morphology of the sample before and after polishing were measured on a ContourGTk-1 3D profile inspection system (Bruker, Billerica, MA, USA).
(1)$$MRR=\frac{\Delta m}{\rho r^{2}\pi t}\times 10^{7}$$
where, Δ*m* is the mass difference before and after polishing, g, *t* is the processing time, min, *ρ* is the density of SiC, g/cm^3^, which is taken as 3.2 g/cm^3^, *r* is the radius of the test sample, mm.

### 2.2. Workpiece Surface Composition Testing

In order to explore the solid-phase chemical reaction between different solid-phase oxidants and SiC and their oxygen production mechanism, the chemical elemental composition of the sample surface before and after polishing was examined by Quanta 200 SEM and the accompanying OXFORDINCA250 energy spectrometer system (FEI, Hillsboro, OR, USA). In addition, the chemical structure composition of the sample surface before and after polishing was examined by Bruker D8 Advance A25 XRD (Bruker, Billerica, MA, USA).

## 3. Analysis of Results

### 3.1. Elements and Content of SiC Surface after Polishing

The percentage of surface oxygen element content on the surface of SiC before and after dry polishing with five solid-phase oxidants was detected by SEM, and the results showed that the initial surface of SiC didnot contain oxygen before polishing. The atomic percentages of surface C and Si are shown in Table 3. After testing, oxygen appeared on the surface of SiC after dry polishing with five solid-phase oxidants.The percentage of oxygen atoms is shown in Figure 3.

The occurrence of the tribochemical reaction of SiC can be reflected by the change of the atomic percentage on its surface, as shown in Figure 3. After polishing with five solid-phase oxidants, although oxygen was produced on the surface, the oxygen atomic percentage content varied, indicating that different degrees and mechanisms of tribochemical reactions occur between the five solid-phase oxidants and SiC. The highest percentage of oxygen atomson the surface reaction layer was observed after the dry polishing of SiC with the solid-phase oxidant Na_2_CO_3_—1.5 H_2_O_2_, and the lowest percentage of oxygen atoms in the surface of SiC was observed after dry polishing with the solid-phase oxidant KClO_3_.

According to the SEM analysis, the appearance of oxygen on the surface of SiC indicated that the solid-phase oxidant could generate oxygen to oxidize the SiC surface under the action of frictional heat to produce an oxidation reaction film on the SiC surface. Therefore, it has been shown that SiC can generate a more shearable reaction film by tribochemical reaction at room temperature [20,21].

### 3.2. Physical Phase Analysis of SiC Surface after Polishing

The XRD results of SiC after dry polishing were compared with those of SiC before dry polishing, as shown in Figure 4. After importing the XRD data before and after dry polishing into Jade software, it wasfound that the same peaks appeared between 30° and 40° and between 70° and 80°. The peak at 32° may be SiO_2_ after software comparison analysis, and the intensity of the detected peak on the surface of the initial SiC was small. No oxygen appears, indicating that the content on the surface of SiC is small and not easy to detect. The intensity of the peak increased after dry polishing, indicating that silicon oxides were generated on the surface.

The appearance and change of some micropeaks in the detection results may be (SiO_2_)_x_, a class of microporous silicate inclusion compounds with a (4,2)-3D structure [22]. In the dry polishing test, a surface tribochemical reaction generated a layered structure.This layered structure may be due to oxidation during the solid-phase oxidant and SiC test period under the thermal effect of tribochemical reaction transformation of oxidation substances.This layered structure includes a multi-functional layer useful for redox, friction reduction, and anti-wear functions [23,24].

### 3.3. Material Removal Rate after Polishing

Figure 5 show the material removal rates of SiC after dry polishing with five different solid-phase oxidants. The results showed that the five different solid-phase oxidants used for the tribochemical mechanical polishing tests all produce material removal from the SiC, indicating that the five solid-phase oxidants used in the tests may have experienced tribochemical reactions with the workpiece material. Among them, the highest material removal rate was achieved with the solid-phase oxidant NaOH and the lowest with the solid-phase oxidant KClO_3_.

The highest material removal rate after dry polishing of SiC with NaOH is due to the tribochemical reaction between SiC and NaOH during the dry polishing process to generate CO and CO_2_ released in the air. On the other hand, because silicon oxides and water-soluble silicates are easily removed by mechanical action, they are also generated on the surface of SiC.

### 3.4. Surface Roughness and Surface Morphology after Polishing

Figure 6 show the changes in surface roughness before and after the dry polishing of SiC using five different solid-phase oxidants. The results showed that the tribochemical polishing tests using five different solid-phase oxidants all affect the surface roughness of the SiC, indicating that the tribochemical interaction between the five solid-phase oxidants used in the tests and the workpiece material all cause the removal of some material from the workpiece surface, thus changing its surface roughness. Among them, the surface roughness of SiC after the action of NaOH increased significantly, while the surface roughness of SiC after the action of other solid-phase oxidants increased slightly but not significantly.

The comparison of SEM before and after the dry polishing of SiC with solid-phase oxidant is shown in Figure 7. The surface of SiC after dry polishing with NaOH showed a significant change in pits and scratches compared to the initial morphology.

## 4. Discussion

### 4.1. Solid-Phase Oxidant Tribochemical Reaction Oxygen Generation Mechanism

From Figure 2, it can be seen that the solid-phase oxidant fills between the SiC and the polishing pad during the polishing process, but under the polishing pressure, the SiC specimen and the polishing pad or solid-phase oxidant can be in contact at the micro-convex body [25]. Friction, local compression, or micro-collisions may occur on the micro-convex body at the polishing interface, which will generate concentrated local stresses at the point of contact (several gigapascals [26])and high flash point temperatures(up to 1000 degrees Celsius [27,28]). Then, under the action of friction and high flash point temperatures, etc., the solid-phase oxidant decomposes oxygen and oxidizes SiC or reacts with SiC by friction chemistry with other media [25,29,30,31].

Sodium percarbonate (Na_2_CO_3_—1.5 H_2_O_2_) is an inorganic substance and white granular solid commonly known as solid H_2_O_2_; it is a strong oxidant. It is easy to separate out oxygen when exposed to moisture to obtain Na_2_CO_3_, H_2_O, and O_2_.In addition, sodium percarbonate is a heat-sensitive substance, dry Na_2_CO_3_—1.5 H_2_O_2_ at 120 °C decomposition. However, in the presence of water, heat, or if mixed with heavy metal and organic material, it is very easy to decompose into Na_2_CO_3_, H_2_O, and O_2_, and its stability decreases with the rise of temperature [32,33]. See Equation (2).
Na_2_CO_3_—1.5 H_2_O_2_(2Na_2_CO_3_—3 H_2_O_2_)→4Na_2_CO_3_ + 6H_2_O + 3O_2_↑ (120 °C)(2)

Studies have shown that the decomposition of sodium percarbonate is an autocatalytic mechanism. In the decomposition of sodium percarbonate, H_2_O is the main catalyst [34]. The product of sodium percarbonate decomposition diffuses to the reaction interface to form intermediates with the reactants, which reduces the activation energy of the reaction and accelerates the reaction. It can be considered that the autocatalytic decomposition of sodium percarbonate proceeds in the following steps.
Na_2_CO_3_—1.5 H_2_O_2_→[Na_2_CO_3_···1.5 H_2_O_2_—H_2_O](3)
[Na_2_CO_3_···1.5 H_2_O_2_—H_2_O]→Na_2_CO_3_ + [1.5 H_2_O_2_—H_2_O](4)
[1.5 H_2_O_2_—H_2_O]→2.5H_2_O + 0.75O_2_↑(5)

The Na_2_CO_3_—1.5 H_2_O_2_ molecule first combines with H_2_O to form the activation complex [Na_2_CO_3_—1.5 H_2_O_2_—H_2_O], which is unstable and quickly decomposes into Na_2_CO_3_. The reactive intermediate [1.5 H_2_O_2_—H_2_O], [1.5 H_2_O_2_—H_2_O] is also unstable and decomposes into H_2_O and O_2_. Where Equation (3) proceeds slowly, Equations (4) and (5) proceed more rapidly and reach an equilibrium quickly [33].

Sodium hydroxide (melting point is 318.4 °C, the boiling point is 1390 °C) powder will turn into molten sodium hydroxide under the action of frictional heat. In addition, the oxidant sodium hydroxide is easily deliquesced in air and reacts with CO_2_ to form Na_2_CO_3_ and H_2_O [35]. See Equation (6).
2NaOH + CO₂→Na₂CO₃ + H₂O(6)

Potassium chlorate (KClO_3_) is an inorganic compound, a colorless or white crystalline powder, that is a strong oxidantandis stable at room temperature. When heated to approximately 360 °C (the melting point of potassium chlorate), oxygen is released, and the reaction mechanism can be expressed in Equation (7). At continuous heating to 610 °C, the rate of oxygen release becomes slower, and the viscosity of the system thickens. At this point, the reaction is as in Equation (8); that is, potassium chlorate is oxidized to potassium perchlorate (KClO_4_) by self-disproportionation. Equations (7) and (8) occur objectively at the same time, and when further heating to 800 °C is conducted, oxygen is released again until the system is completely changed to potassium chloride, such as Equation (9) [36].
2KClO_3_→2KCl + 3O_2_↑ (365 ± 5 °C)(7)
KClO_3_→KClO_4_ + KCl(8)
KClO_4_→KCl + O_2_↑(9)

Potassium permanganate (KMnO_4_) is a strong oxidant with a melting point of 240 °C. Its thermal decomposition process is complex, and within 190–700 °C, the following decomposition reactions are produced [37,38,39].
6KMnO_4_→2K_2_MnO_4_ + K_2_Mn_4_O_8_ + 4O_2_↑(10)
2KMnO_4_→KMnO_2_ + O_2_↑(11)
3K_2_MnO_4_→2K_3_MnO_4_ + MnO_2_ + O_2_↑(12)

In addition, light has a catalytic effect on the decomposition of potassium permanganate, KMnO_4_ is not very stable in sunlight, and KMnO_4_ can spontaneously undergo redox reactions with H_2_O [40].
4KMnO_4_+ 2H_2_O→4KOH + 4MnO_2_ + 3O_2_↑(13)

Potassium iodate (KIO_3_) is an inorganic substance. It is a colorless crystal, and its melting point is 560 °C (decomposition). It can be decomposed into KI by heat; KI reacts with O_2_ and H_2_O in moist air to form KOH [41,42].
2KIO_3_→2KI + 3O_2_↑ (525 °C)(14)
4KI + O_2_ + 2H_2_O→2I_2_ + 4KOH(15)

### 4.2. Mechanism of Tribochemical Oxidation Reaction on the Surface of SiC

(1)Solid-phase oxidant NaOH

See Equation (6), the solid-phase oxidant NaOH readily deliquesces in air and reacts with CO_2_ to form Na_2_CO_3_ and H_2_O [35]. In addition, under the action of friction, the suspended silicon bond in SiC will also undergo the following tribochemical reaction. The main chemical equation is as follows [42,43]:Si+2NaOH+H_2_O→Na_2_SiO_3_ + 2H_2_↑(16)

The Na_2_SiO_3_ produced by the reaction is soluble and can be easily removed from the SiC surface by mechanical action.

(2)Other solid-phase oxidants

From Section 3.1, Na_2_CO_3_—1.5 H_2_O_2_, KIO_3_, KClO_3_, and KMnO_4_ can produce O_2_ by decomposition under the action of friction heat, then the surface of the SiC undergoes a tribochemical oxidation reaction under the action of frictional heat and other media [20,21].
SiC +2O_2_→SiO_2_ + CO_2_↑(17)

(3)Tribochemical hydration reaction on SiC surface

During the polishing process, due to the high flash point temperature at the polishing interface, SiC reacts with H_2_O to produce SiO_2_ on the SiC surface. The main chemical reactions are as follows [44,45,46,47]:SiC +2H_2_O→SiO_2_ + C+2H_2_↑(18)
SiC +4H_2_O→SiO_2_ + CO_2_ + 4H_2_↑(19)
SiC +O_2_ + H_2_O→SiO_2_ + CO↑+H_2_↑(20)

The flash point temperature during polishing excites the oxidation reaction in Equations (18)–(20). Therefore, the temperature of the test environment is so low that it does not affect the occurrence of thetribochemical reaction and has little effect on friction behavior [48,49].

Thus, as described above, the surface of SiC transforms into SiO_2,_ Na_2_SiO_3,_ or a surface film composed of SiO_2_ and Na_2_SiO_3_ [26,42,43]. The resulting product, regardless of the state in which the generated product exists, is less hard than SiC. This oxide layer is easily removed using abrasive.

### 4.3. Material Removal Mechanism of Solid-Phase Oxidant

From Figure 3 and Figure 4, after polishing SiC with five solid-phase oxidants, it was found that the surface of SiC contained oxygen, and the surface products of SiC are SiO_2_ and silicon oxides. This illustrates the complex tribochemical reactions generated at the polishing interface during the polishing process, see Equations (2)–(20).Moreover, SiO_2_ on the surface of SiC is obtained by the tribochemical reaction shown in Equations (17)–(20).

#### 4.3.1. Solid-Phase Oxidant Sodium Hydroxide (NaOH)

(1)The surface phases of SiC after polishing all contain oxygen, and the surface compounds are SiO_2_ and silicon oxides. On the one hand, the solid-phase oxidant NaOH reacts with CO_2_ in the air under the action of friction to form Na_2_CO_3_ and H_2_O, see Equation (6).On the other hand, the SiC reacts with H_2_O in the air to form SiO_2_ on its surface, see Equations (18)–(20).The CO and CO_2_ generated by the reaction escape into the air, and the generated C is removed by friction; the SiO_2_ generated is attached to the SiC surface. Figure 8 and Figure 9 show the surface morphology of SiC after dry polishing with NaOH.(2)Under the action of friction, the suspended silicon bonds of SiC also undergo a tribochemical reaction with NaOH, see Equation (16). The reaction produces soluble Na_2_SiO_3_, which is removed from the SiC surface. As can be seen from the SEM inspection of the enlarged Figure 7c and Figure 9b, after polishing, more small pits appear on the SiC surface with a smoother edge, not just brittle fracture removal. It can be shown that the Na_2_SiO_3_ produced by the reaction is removed by dissolution.

#### 4.3.2. Solid-Phase Oxidant Sodium Percarbonate (Na_2_CO_3_—1.5 H_2_O_2_)

The solid-phase oxidant Na_2_CO_3_—1.5 H_2_O_2_ decomposes into Na_2_CO_3_, H_2_O, and O_2_ under the action of friction, see Equation (2), and the products, in turn, produce a tribochemical reaction with SiC, as shown in Equations (17)–(20).The generated SiO_2_ adheres to the surface of SiC, the generated CO and CO_2_ escape into the air, and the generated C is removed by friction.The material removal rate consists of the products C, CO, and CO_2_; however, the Si atoms in the SiC are not lost, but partly oxidized to SiO_2_.Therefore, the removal rate is lower than that of the solid-phase oxidant NaOH, see Figure 5.

#### 4.3.3. Solid-Phase Oxidant Sodium Iodate (KIO_3_)

Under frictional heat, KIO_3_ produces O_2_ and generates KI, see Equation (14), which in turn reacts with the SiC surface in an oxidation reaction, see Equation (17). Furthermore, KOH is generated from the reaction of KI with O_2_ and H_2_O in the air, which can also provide an alkaline environment to induce the SiC to react with O_2_, see Equation (15). Since there is no H_2_O in the KIO_3_ decomposition reaction equation, SiC may also react with H_2_O in the air by frictional chemistry under the action of frictional heat, see Equations (18)–(20). However, the Si atoms in SiC are also not lost but partially oxidized to SiO_2_.

The SiO_2_ generated by the tribochemical reaction adheres to the SiC surface, and the generated gas escapes into the air, thus creating a material removal rate.

#### 4.3.4. Solid-Phase Oxidant Sodium Chlorate (KClO_3_)

Under frictional heat, KClO_3_ decomposes to produce O_2_ and generates KCl, see Equations (7)–(9).The resulting O_2_ reacts with the SiC surface in an oxidation reaction, see Equation (17). The SiO_2_ generated by the tribochemical reaction adheres to the SiC surface, and the generated CO_2_ escapes into the air, thus creating a material removal rate.

Since there is no H_2_O in the KCIO_3_ decomposition reaction equation, SiC may also react with H_2_O in the air by frictional chemistry under the effect of frictional heat, see Equations (18)–(20). However, the Si atoms in the SiC are not lost, but partially oxidized to SiO_2_.

#### 4.3.5. Solid-Phase Oxidant Sodium Permanganate (KMnO_4_)

Under the action of frictional heat, KMnO_4_ decomposes to produce O_2_ and generates KCI, see Equations (10)–(12). In turn, this reacts with the SiC surface in an oxidation reaction, see Equation (17). At the same time, KMnO_4_ can spontaneously react with H_2_O in the air to produce O_2_ via a redox reaction.

The SiO_2_ generated by the tribochemical reaction adheres to the SiC surface, and the generated CO_2_ escapes into the air, thus creating a material removal rate.

Since there is no H_2_O in the KMnO_4_ decomposition reaction equation, SiC may also produce a tribochemical reaction with H_2_O in the air under the effect of friction heat, see Equations (18)–(20). However, Si atoms in SiC are not lost, but partly oxidized to SiO_2_.

To sum up, under the action of friction, a more complex tribochemical reaction was produced between the solid-phase oxidant NaOH and SiC and air medium, not only the removal of C atoms, but also Si atoms. Comparatively, in the other reactions, only the C atoms were removed. Therefore, the largest removal rate wasproduced when polishing with the solid-phase oxidant NaOH, see Figure 5. In addition, since Na_2_SiO_3_ was produced when the solid-phase oxidant NaOH was used for polishing, and Na_2_SiO_3_ was easily removed by dissolution, more small pits with a depth of 1400 nm were produced on the surface, see Figure 9. As a result, the polishing surface roughness was also at a maximum when the solid-phase oxidant NaOH was used, see Figure 6.

## 5. Conclusions

(1)After dry polishing SiC with all five solid-phase oxidants, oxygen was detected on the surface, but the percentage of oxygen atoms on the surface after polishing varied. The highest percentage of oxygen atoms was observed after dry polishing SiC with Na_2_CO_3_—1.5 H_2_O_2_ and the lowest percentage of oxygen atoms was observed on the surface after dry polishing with KClO_3_.(2)From the XRD results, it couldbe seen that the appearance of surface oxygen was due to the tribochemical reaction between the five solid-phase oxidants and the SiC in the polishing process. The reaction product was known to be silicon oxides, and the main substance was SiO_2_. In addition, under the action of friction, due to the high flash point temperature at the polishing interface, SiC reacted with H_2_O and generated a SiO_2_ oxide layer on the SiC surface.(3)The material removal rate was calculated by measuring the mass before and after polishing, and the highest material removal rate couldbe obtained after dry polishing of SiC with NaOH and the lowest material removal rate could be obtained after dry polishing with KClO_3_.(4)After polishing SiC with oxidant NaOH, soluble Na_2_SiO_3_ was generated. Therefore, more obvious scratches and pits appeared on the surface of SiC, and the roughness hada substantial increase.The surface roughness of the remaining four solid-phase oxidants didnot change significantly after polishing.

## Figures and Tables

**Figure 1 micromachines-12-01547-f001:**
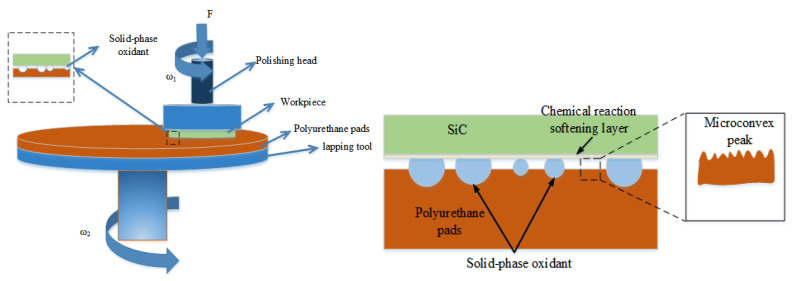
The schematic of tribochemica lmechanical polishing.

**Figure 2 micromachines-12-01547-f002:**
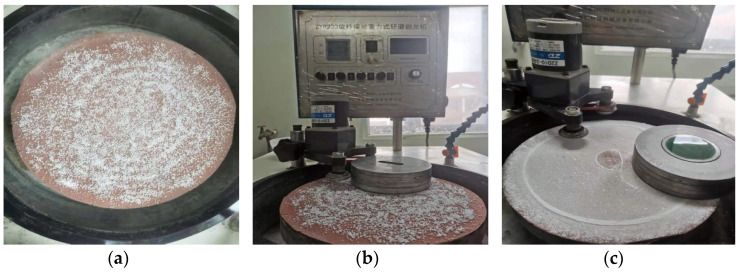
Test process. (**a**) The oxidant is evenly distributed on the polishing pad. (**b**) Start polishing. (**c**) Finish polishing.

**Figure 3 micromachines-12-01547-f003:**
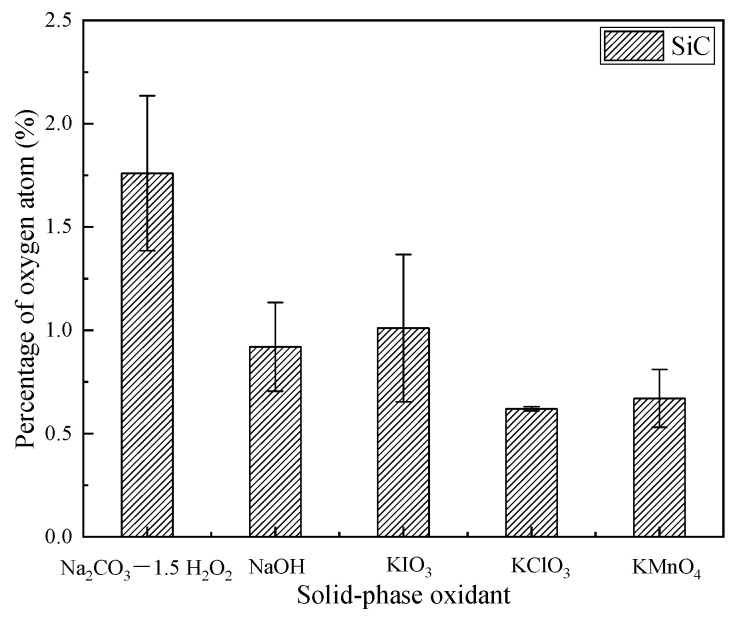
Percentage of oxygen atoms on SiC surface after dry polishing.

**Figure 4 micromachines-12-01547-f004:**
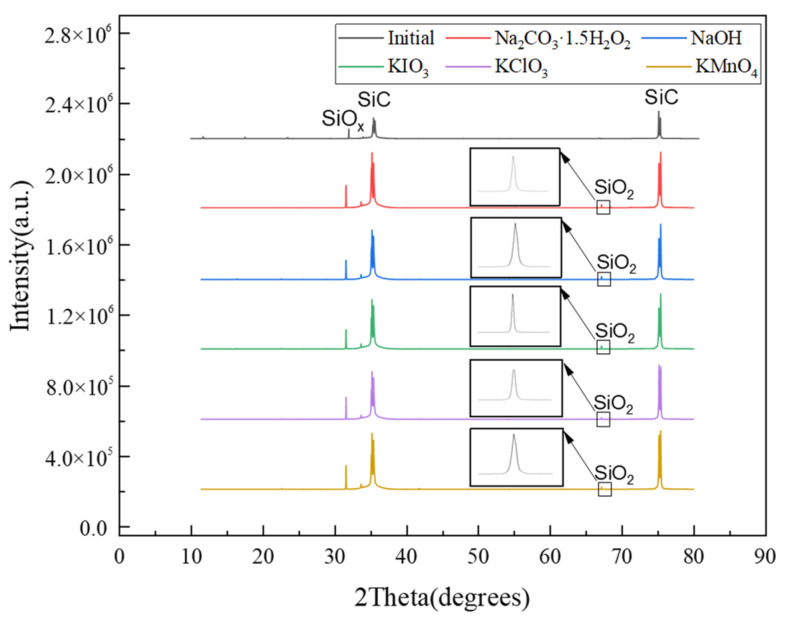
XRD of SiC after dry polishing with solid-phase oxidant.

**Figure 5 micromachines-12-01547-f005:**
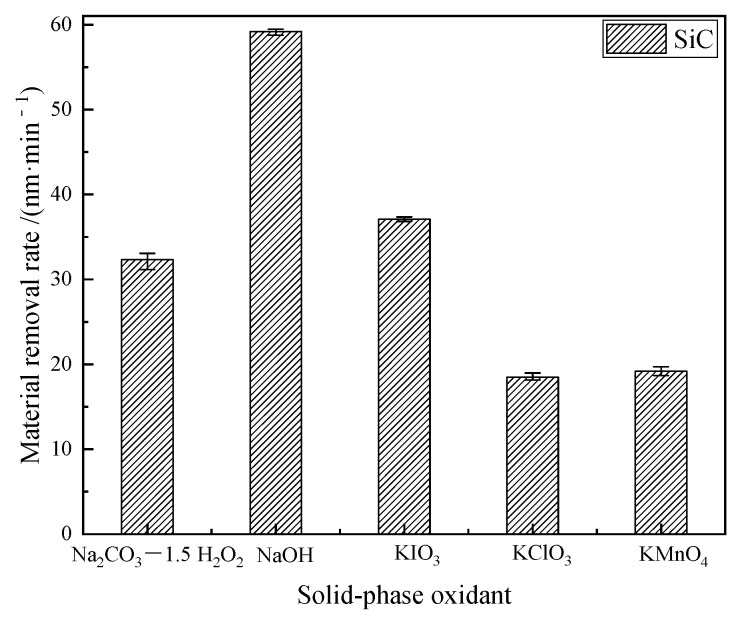
Material removal rate after dry polishing.

**Figure 6 micromachines-12-01547-f006:**
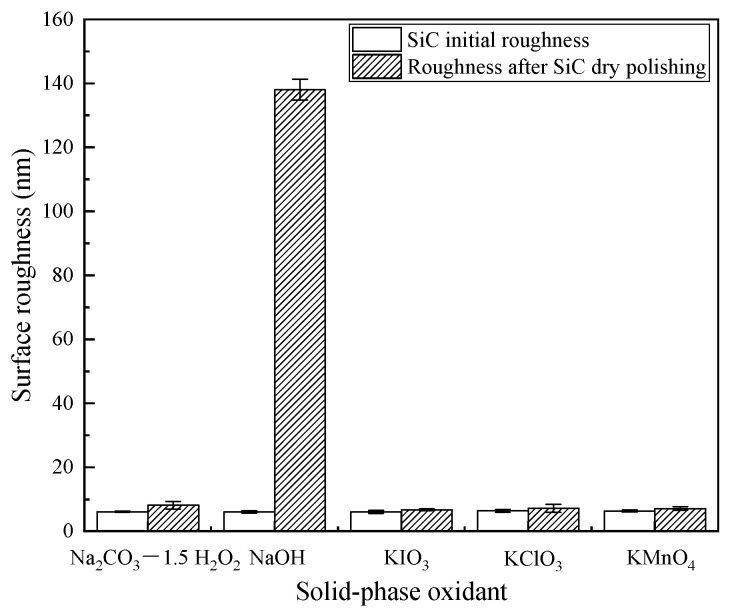
Surface roughness.

**Figure 7 micromachines-12-01547-f007:**
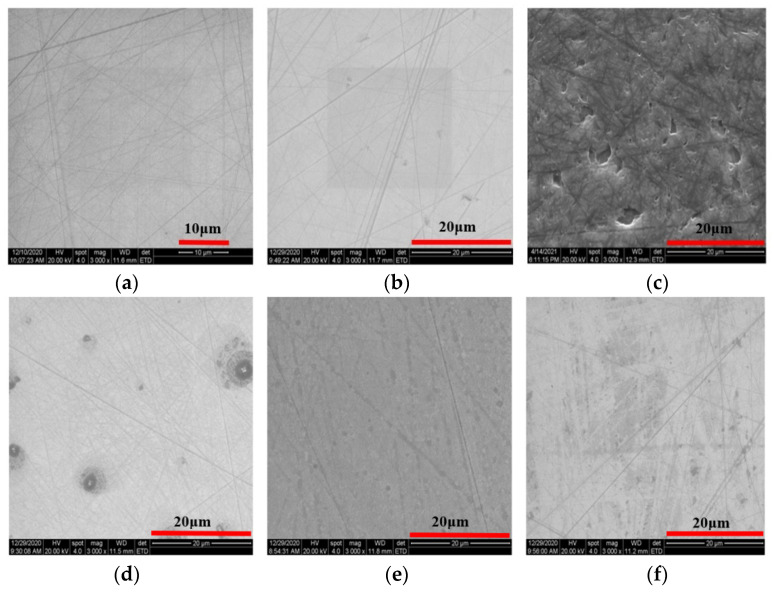
SEM of SiC surface after dry polishing with solid-phase oxidant. (**a**) Initial. (**b**) Na_2_CO_3_—1.5 H_2_O_2_. (**c**) NaOH. (**d**) KIO_3_. (**e**) KClO_3_. (**f**) KMnO_4_.

**Figure 8 micromachines-12-01547-f008:**
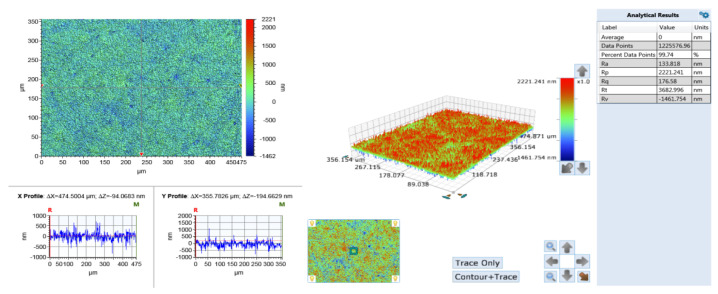
Surface morphology after dry polishing with NaOH.

**Figure 9 micromachines-12-01547-f009:**
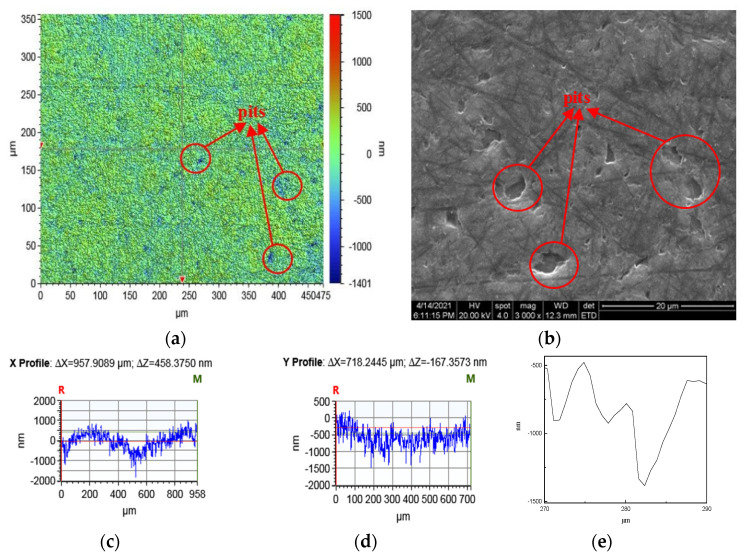
Surface pits after dry polishing with NaOH. (**a**) White light interferometer detection of surface pits. (**b**) SEM inspection of surface pits. (**c**) Partial X coordinate shape. (**d**) Local Y coordinate shape. (**e**) Local enlargement of the pit.

**Table 1 micromachines-12-01547-t001:** Polishing process parameters.

Factors	Speed of Polishing Tool n_1_ (r/min)	Speed of Polishing Head n_2_ (r/min)	Polishing Pressure P (psi)	Time t (h)
Parameters	60	45	2	1.5

**Table 2 micromachines-12-01547-t002:** Solid-phase oxidant and its composition used in the test.

	1	2	3	4	5
6H-SiC	Na_2_CO_3_—1.5 H_2_O_2_	NaOH	KIO_3_	KClO_3_	KMnO_4_

**Table 3 micromachines-12-01547-t003:** Initial surface elements and content of SiC.

Element	C (%)	Si (%)
Atomic percentage of SiC surface elements	45.67	54.33

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
