# Peer review of "Study on the Mechanism of Solid-Phase Oxidant Action in Tribochemical Mechanical Polishing of SiC Single Crystal Substrate"

_micromachines, 2021, doi:10.3390/mi12121547_

Round 1
Reviewer 1 Report
Review comments
- Please simply, the abstract in a concise manner.
- Please modify the sentences in such a manner that the meaning becomes clear to the reader, check the very first sentence of the abstract section.
- Please check your grammar throughout the manuscript, use past tense to describe your article story.
- Make your article like a story, to be interesting for the readers.
- Avoid repetition of the same descriptions throughout the manuscript.
- Please make clear graphs using high fonts for text, thicker lines and using scientific notations (400000 can be written as 4*105 )
- Please provide reference for your reactions in equ 3, 4,5, and 6.
- Please repeat your manuscript and concise your text by removing extra descriptors and using compact sentences, it looks very awkward now.
Reviewer 2 Report
The work is well presented and commented with appropriate supporting information. before publishing a spell check is required (too many typos, some are highlighted in the attached file)

Reviewer 3 Report
Chemical mechanical polishing (CMP) is an important technique for obtaining ultra-smooth, damage-free surfaces. Na2CO3-1.5, H2O2,KClO3, KMnO4, KIO3, and NaOH were selected for dry polishing tests with 6H-SiC single crystal substrate. Authors report that after the polishing with the five solid-phase oxidants, oxygen element was found on the surface of SiC single crystal substrate, indicating that all the five solid-phase oxidants can have complex tribochemical reactions with SiC single crystal substrate, and their reaction products are mainly SiO2 and (SiO2)x compounds. All the results and meachnisms are interesting, espcially that SiC is very important material in electronics and hi-tech industry.
The manuscript suits to Special Issue: "Advances in Ultra-Precision Machining Technology and Applications". Therefore, I think that manuscript can be accepted after minor revision:
- Table 3 and Table 4 are not critical and should be moved from the manuscript to ESI.
- Figure 4 - the quality of it has to be improved.
- Figure 5 - can Authors assed the error (SD) for material removal rate?
- Please check all manuscript: grammar should be verified, as well as all typos etc.
Round 2
Reviewer 1 Report
accepted in the current